# Hydrophilic Modification of Polyester/Polyamide 6 Hollow Segmented Pie Microfiber Nonwovens by UV/TiO_2_/H_2_O_2_

**DOI:** 10.3390/molecules28093826

**Published:** 2023-04-30

**Authors:** Baobao Zhao, Xu Han, Chenggong Hu, Xiaoming Qian, Yongchao Duo, Zhen Wang, Quan Feng, Quan Yang, Dongxu Han

**Affiliations:** 1School of Textile and Garment, Anhui Polytechnic University, Wuhu 241000, China; zhaobaobao@ahpu.edu.cn (B.Z.); hanxu@ahpu.edu.cn (X.H.); yangquanahpu@163.com (Q.Y.); handongxuahpu@163.com (D.H.); 2Advanced Fiber Materials Engineering Research Center of Anhui Province, Anhui Polytechnic University, Wuhu 241000, China; 3School of Chemical and Environmental Engineering, Anhui Polytechnic University, Wuhu 241000, China; huchenggong2014@163.com; 4School of Textile Science and Engineering, Tiangong University, Tianjin 300387, China; qxmtjpu@163.com (X.Q.); 18222826909@163.com (Y.D.)

**Keywords:** microfiber nonwovens, hydrophilic modification, splittable filament, titanium dioxide, water vapor transmission rate, water absorption

## Abstract

Polyester/polyamide 6 hollow segmented pie bicomponent spunbond hydro-entangled microfiber nonwovens (PET/PA6) with a microfilament structure have recently emerged in many markets around the world due to their green, high-strength, and lightweight properties. However, PET/PA6 is highly hydrophobic, which inhibits its large-scale application at present. In order to enhance the hydrophilic performance of PET/PA6, many methods have been applied, but the effects are not obvious. Ultraviolet (UV) irradiation treatment has proven to be an effective method to improve the hydrophilicity of fabrics. Herein, the aim of this paper was to investigate hydrophilic modification of PET/PA6 by UV/TiO_2_/H_2_O_2_. The effect of H_2_O_2_, nano-TiO_2_, and UV irradiation time on the morphology, elemental composition, hydrophilic properties, and mechanical properties of PET/PA6 were systematically investigated. The results showed that the modified microfibers were coated with a layer of granular material on the surface. It was found that the C 1*s* peak could be deconvoluted into six components (C–C–C, C–C–O, O–C=O, N–C=O, N–C–C, and C–C=O), and a suitable mechanism was proposed. Moreover, the water contact angle of PET/PA6 modified by 90 min irradiation with UV/TiO_2_/H_2_O_2_ decreased to zero in 0.015 s, leading to the water vapor transmission rate and the water absorption reaching 5567.49 g/(m^2^·24 h) and 438.81%, respectively. In addition, the modified PET/PA6 had an excellent liquid wicking height of 141.87 mm and liquid wicking rate of 28.37 mm/min.

## 1. Introduction

Polyester/polyamide 6 hollow segmented pie bicomponent spunbond hydro-entangled microfiber nonwovens (PET/PA6), as splittable filaments have shown superior properties such as softness, higher absorbency due to a higher surface area, and higher filtration efficiency [1,2,3]. Nevertheless, the main component of microfibers leads to high hydrophobicity (absorbing ~0.4–4.5% of moisture), limiting their further applications [4,5,6]. Therefore, hydrophilic modification of PET/PA6 is an attractive approach to solving this problem.

Recently, many methods and techniques, such as acid treatments and micro/nanostructures, have been employed to improve the hydrophilic wettability of PET/PA6. Xu et al. [7] initially fabricated PET/PA6 and then explored its modification with ethylenediamine and amino acids. The results showed that amino-acid-modified PET/PA6 had decent water flux and adsorption performance. However, the process of acid treatment affected the mechanical properties of PET/PA6. The fineness of the fiber affects the surface hydrophilicity of nonwovens. In recent years, many researchers have introduced hierarchical micro/nanostructures into PET/PA6 to obtain super-hydrophilic nonwovens. Duo et al. [8] systematically investigated the influence of polyacrylonitrile (PAN) nanofiber fineness on the performance of PET/PA6, and found that the water vapor transmission rate and moisture absorption performance increased, while the diameter of the nanofibers decreased from 950 nm to 200 nm. Moreover, the effects of polyhydroxybutyrate (PHB) nanofiber content with an average diameter of 400 nm on the structure and properties of PET/PA6 were investigated [9]. The results indicated that the use of PHB nanofiber with a large number of carboxyl groups increased sanitary performance. As the PHB nanofiber content increased from 0% to 20%, the water contact angle decreased from 111.64° to 59.31°, resulting in a 44% increase in water vapor transmission rate and 22.3% increase in moisture absorption. Zhao et al. [10] analyzed the effects of thermoplastic polyurethane (TPU)/sulfonated polysulfone (SPSf) nanofiber with an average diameter of 120 nm on the hydrophilicity of PET/PA6, and they concluded that the addition of TPU/SPSf nanofiber to PET/PA6 could enhance the water vapor transmission rate and moisture absorption. However, the use of the abovementioned nanofiber is known to be difficult due to factors such as low fiber strength and low production efficiency. 

Ultraviolet (UV) irradiation treatment, as a relatively simple and straightforward strategy, has been used for the surface modification of fabrics to increase hydrophilicity. Liang et al. [11] experimentally modified the PET fabric surface with UV irradiation and titanium dioxide nanoparticles (nano-TiO_2_). The results demonstrated that the water contact angle of PET fabric modified with 40 g/L nano-TiO_2_ and 60 min UV irradiation decreased to zero in 4.11 s, while the water absorption increased to 89.86%. Moreover, hydrogen peroxide (H_2_O_2_) addition reduced the disappearance time of water droplets to 3.29 s [12]. Li et al. [13] further found that the water contact angle of modified PET fabric, in the best conditions (3% nano-TiO_2_, 5% H_2_O_2_, and 3% sodium hydroxide (NaOH), under 1000 W UV irradiation for 60 min), decreased to zero in 3 s, while the water absorption increased from 85% to 104%. At the same time, Zhang et al. [14] also reported that PET fabrics treated using a concentration of 4% H_2_O_2_ and a concentration of 1% nano-TiO_2_ under UV irradiation for 80 min could became super-hydrophilic. However, so far, very little research has been conducted on the application of UV irradiation treatment to filament microfiber nonwovens. 

In this study, experiments were designed to explore the super-hydrophilic modification of PET/PA6 using nano-TiO_2_ as a photocatalyst in the presence of H_2_O_2_ under UV irradiation. The effects of H_2_O_2_, nano-TiO_2_, and UV irradiation time on the performance of PET/PA6 were systematically investigated. This study had the goal of developing a method to assist in solving the current hydrophobicity problem of polyester/polyamide 6 hollow segmented pie microfiber nonwovens. These results are also useful for future hydrophilic modification of other filament microfiber nonwovens. 

## 2. Results and Discussion

### 2.1. Morphology Analysis

Figure 1 shows the SEM images of fabricated PET/PA6 and its surface EDS spectrum. As shown in Figure 1(a1,a2), the surface of untreated microfiber was relatively smooth and shiny. The microfiber diameter was 5.89 um. The surface roughness of the microfibers after modification with UV/H_2_O_2_ slightly increased (see Figure 1(b1,b2). Some bright spots could be seen under high magnifications, possibly due to the reaction of H_2_O_2_ and amide bond, which generated cracking of the PA6 macromolecular chain, or due to flaws generated during manufacturing [15,16]. When TiO_2_ was introduced, the modified microfibers were coated with a layer of granular material on the surface (see Figure 1(c1,c2,d1,d2,e1,e2)), which could not be removed after repeated washing of the materials. The microfiber diameters were 6.16 μm, 6.23 μm, and 6.55 μm, respectively. High-magnification electron micrographs showed that many irregular fine particles were distributed on the surface of the treated microfibers. The partial agglomeration of particles caused some large micron-sized particles to adhere to the fiber surface. 

In addition, EDS was used to characterize the surface elements of the microfibers, and the results are shown in Figure 1(a3,b3,c3,d3,e3). It can be seen that only C and O elements were present in specimen M1 and specimen M2, while Ti elements were present in specimen M3, specimen M4, and specimen M5. The weight concentration and atomic concentration of Ti on the surface of nonwovens modified with UV/TiO_2_ were 3.99% and 1.14%, respectively, while the weight concentration and atomic concentration of Ti on the surface of nonwovens modified with UV/TiO_2_/H_2_O_2_ reached 3.63% and 1.03%. After the joint UV/TiO_2_/H_2_O_2_ action, nano-TiO_2_ photocatalysis resulted in a significant enhancement of the microfiber surface oxidation. Moreover, by increasing the UV irradiation time, the weight concentration and atomic concentration of Ti further increased to 7.65% and 2.25%. A TiO_2_ network was formed on the surface of filaments via the interaction of the TiO_2_ sol with OH groups [17]. With increasing UV irradiation time, accumulation on the TiO_2_ layer increased; that is, the number of TiO_2_ particles increased [18]. Figure 1 also shows the AFM images of specimens M3, M4, and M5, indicating roughness of 339 nm, 641 nm, and 1007 nm; hence, the surfaces became much rougher upon depositing the TiO_2_ layer. 

### 2.2. XPS Analysis

In order to further analyze the changes in surface elemental composition of fabricated PET/PA6, X-ray photoelectron spectroscopy (XPS) was carried out. The surface composition was determined, and the results are shown in Figure 2 and Table 1. Figure 2a shows the wide survey scan of XPS spectra in the range 0–1200 eV. The survey spectra of M1 and M2 samples were dominated by the signals of C, O, and N. The spectra of M3, M4, and M5 samples displayed signals of C, O, N, and Ti. The most significant change in the spectra was the appearance of a new peak, corresponding to a titanium atom, at a chemical shift of 458 eV. The Ti content was found to increase from 7.47% to 9.34% (Table 1). At the same time, the appearance of the new component was accompanied by a reduction in the amount of C from 78.80% to 55.50% as more and more TiO_2_ was introduced to the microfiber surface. 

Furthermore, the detailed XPS spectra of the C 1*s* region for M1–M5 samples are shown in Figure 2b–f. Analyzing the C 1*s* core line acquired on the M1 surface (Figure 2b), it was found that the peak could be deconvoluted into five components. The first component of C 1*s* at lower BE (284.2 eV) was due to the C–C–C bond, while the second one at 286.2 eV was due to the C–C–O bond. The third component located at 288.5 eV was likely due to the O–C=O bond. The fourth component and the fifth component (around 287.7 eV and 285.4 eV) originated from the N–C=O and the N–C–C bonds, respectively. According to a comparison of the carbon spectra (binding energy of C1 s*)* in Figure 2c–f, a component at 286.9 eV was additionally observed, possibly indicating the presence of C–C=O. 

The mechanism of PET and PA6 degradation has been discussed in the literature. The main degradation processes of PET are known to be the Norrish type I pathway reported by Fechine et al. [19] and Claire et al. [20], and the Norrish type II reactions proposed by Day and Wiles [21]. The degradation mechanism of PA6 occurs via the amido bond, considered to be the weakest bond in the PA6 molecule, as discussed by Li et al., who indicated a series of degradation reactions initiated by a mercury lamp [22]. The proportion of carbon atoms in the different bonding environments is shown in Table 1. It can be seen that the prportion of C–C–C and N–C=O bonds observed decreased, while the proportion of C–C–O, O–C=O, C–C=O, and N–C–C bonds observed increased. Furthermore, the proportion of C–C=O bonds increased from 0% to 6.04%, indicating aldehyde formation due to chain scission [20]. On the basis of XPS data, the mechanism for hydrophilic modification of PET/PA6 by UV/TiO_2_/H_2_O_2_ is proposed in Figure 3, in agreement with earlier work by Claire et al. and Li et al.

### 2.3. Hydrophilic Properties

In addition, in order to further confirm the effect of hydrophilic modification, the contact angle, water vapor transmission rate, water absorption, capillary effect, and waterproof performance of PET/PA6 were characterized, as shown in Table 2 and Figure 4. As can be seen from Table 2, the contact angle of the unmodified of PET/PA6 (M1) was 137.718° upon initial contact with water droplets and 137.062° at 0.015 s of contact; the droplet disappearance time (when the contact angle was 0°) was 27.465 s. The PET/PA6 nonwovens were impregnated with H_2_O_2_ aqueous solution (M2), nano-TiO_2_ hydrosol (M3), and nano-TiO_2_/H_2_O_2_ hydrosol (M4), and then irradiated using a UV lamp for 45 min. The contact angles of M2, M3, and M4 upon first contact with water droplets were 136.734°, 134.637°, and 134.637°, whereas those at 0.015 s contact were 136.389°, 133.150°, and 132.769°, respectively. The disappearance times of water droplet of M2, M3, and M4 were 1.680 s, 0.271 s, and 0.072 s, respectively. It can be seen that the modified PET/PA6 fabric became super-hydrophilic (i.e., a contact angle <10° within 10 s). Moreover, when the time of UV irradiation was increased to 90 min (M5), the contact angle of water droplets rapidly changed to 0° at 0.015 s on the surface of the modified microfiber nonwovens. This indicates that the efficiency of UV modification of hollow segmented pie bicomponent spunbond hydro-entangled microfiber nonwovens was substantially improved by the introduction of H_2_O_2_ and TiO_2_.

Figure 4 shows the water vapor transmission rate, water absorption, liquid wicking height, liquid wicking rate, and spray rating of filament microfiber nonwovens before and after modification. As shown in Figure 4a, the water vapor transmission rate of the unmodified hollow segmented pie bicomponent spunbond hydro-entangled microfiber nonwovens was 3820.56 g/(m^2^·24 h), which was improved by modification with UV/H_2_O_2_, UV/TiO_2_, and UV/TiO_2_/H_2_O_2_. After 90 min of UV irradiation (M5), the water vapor transmission rate of the treated nonwovens reached 5567.49 g/(m^2^·24 h). From Figure 4b, it can be seen that the changes in water absorption of nonwovens before and after modification showed the same pattern as the water vapor transmission rate, reaching as high as 438.81% upon modification by UV/TiO_2_/H_2_O_2_ for 90 min. Figure 4c shows the capillary effect of fabricated filament microfiber nonwovens. As can be seen from the figure, the liquid wicking height and liquid wicking rate of the unmodified nonwoven were only 52.83 mm and 10.57 mm/min. When the nonwovens were modified by UV/H_2_O_2_, UV/TiO_2_, and UV/TiO_2_/H_2_O_2_ in turn, both the liquid wicking height and the liquid wicking rate gradually increased. Moreover, as the modification time of UV irradiation increased to 90 min, the liquid wicking height and liquid wicking rate of the nonwovens further increased, reaching 141.87 mm and 28.37 mm/min, respectively. Therefore, the moisture permeability, water absorption, and capillary effect all indicated that the UV/TiO_2_/H_2_O_2_ modification greatly improved the hydrophilicity of PET/PA6. In addition, it can be seen from Figure 4d that the PET/PA6 before and after modification had excellent waterproof performance with a spray rating of level 1, indicating complete wetting of the whole upper surface.

### 2.4. Mechanical Properties

Figure 5 shows the mechanical properties of PET/PA6 before and after modification. It can be seen from Figure 5a that the tensile strength of modified nonwoven by UV/H_2_O_2_ decreased, while the elongation at break increased. The reason may be that the UV radiation destroyed the fiber structure, increased the fiber crystallinity, and increased the brittleness. The tensile strength of the nonwovens after UV/TiO_2_ modification for 45 min and UV/TiO_2_/H_2_O_2_ modification for 45 min increased in both cases, but the difference was not significant. This is because the nano-TiO_2_ sol can be used as a UV shielding agent and has a protective effect on the internal structure of fibers [23,24]. However, as the irradiation time increased to 90 min, the tensile strength decreased again, due to the high-energy UV radiation penetrating the TiO_2_ protective film layer and having an effect on the fiber structure. In addition, it can be seen from Figure 5b that the softness of the nonwoven modified by UV/H_2_O_2_ did not change much after modification. When nano-TiO_2_ sol was introduced, the softness of the nonwovens increased significantly. This is related to the thickness of the TiO_2_ film layer on the fiber surface. The mechanical properties of the nonwovens before and after the modification can meet the demands of nonwoven textiles for synthetic leather [2]. Furthermore, as shown in Figure 5c, PET/PA6 nonwovens before and after modification had different crease recovery angles. The crease recovery property of unmodified nonwovens was greater than that of modified nonwovens. Modified nonwovens were easily wrinkled, with the recovery of flatness after being wrinkled being difficult. 

## 3. Materials and Methods

### 3.1. Materials

Polyester (PET, FC510) chips with a density of 1.38 g/cm^3^ were purchased from Yizheng Chemical Fiber Co., Ltd. (Yizheng, China). Nylon 6 (PA6, 1013B) chips with a density of 1.15 g/cm^3^ were obtained from Ube Co., Ltd. (Dongjing, Japan). Hydrogen peroxide solution (H_2_O_2_, 30%) and ethanol (99.5%) were acquired from Shanghai Macklin Biochemical Co., Ltd. (Shanghai, China). Nano titanium dioxide sol (nano-TiO_2_ sol, 17%, particle size 10–20 nm) was supplied by Nanjing XFNANO Materials Tech Co., Ltd. (Nanjing, China). The polyurethane sponge (40 D) was purchased from Jinhua Yuechang Trading Co., Ltd. (Jinhua, China). Deionized water with the conductivity of <5 μs/cm was used in all experiments.

### 3.2. Preparation and Modification of PET/PA6

Figure 6 shows the preparation and hydrophilic modification process of PET/PA6 hollow segmented pie bicomponent spunbond hydro-entangled microfiber nonwovens (PET/PA6). Firstly, the PET/PA6 nonwovens were produced using bicomponent spunbond and hydro-entanglement technology [2,25]. Secondly, the filament microfiber nonwovens were washed with ethanol and then immersed in individual impregnating solutions. A detailed description of the impregnating solution and UV irradiation time is provided in Table 3. Thirdly, the treated PET/PA6 nonwovens were arranged on a water-filled polyurethane sponge and irradiated using a high-intensity UV lamp (24 W/cm^2^ emitting at 254 nm) for a certain period of time (45 min or 90 min). The constant distance of UV irradiation was 30 cm. Lastly, the samples were dried in an oven for 2 h under a high temperature (80 °C) to obtain the modified PET/PA6. 

### 3.3. Characterization of PET/PA6

#### 3.3.1. Morphology

The surface morphology and elemental composition of fabricated PET/PA6 were characterized using a scanning electron microscope (SEM, S4800, Hitachi, Japan) equipped with an energy-dispersive spectrometer (EDS) system. In order to enhance electric conductivity, the samples were sputtered with gold before SEM measurement. The fiber diameter and distribution of fabricated PET/PA6 were measured using Adobe Photoshop CS6 image analysis software (MacOSX 10.7) [10]. The mean surface roughness (*R*_a_, nm) was carried out using AFM (SPA400, Seiko, Japan) in tapping mode at room temperature in air. 

#### 3.3.2. XPS

The surface chemical composition of fabricated PET/PA6 was tested by X-ray photoelectron spectroscopy (XPS, Thermo Scientific K-Alpha, Waltham, MA, USA). 

#### 3.3.3. Contact Angle

The surface wettability of fabricated PET/PA6 was determined using a contact angle meter (OCA20, Dataphysics, Stuttgart, Germany) equipped with video capture in accordance with the DB44/T 1872-2016 standard. The test droplet was 3 uL. The contact angle was tested when the droplet touched the sample surface at 0 s and 0.015 s, and the droplet complete absorption time (i.e., at a contact angle of 0°, denoting the droplet disappearance time) was recorded. At least 10 measurements were averaged to obtain a reliable value.

#### 3.3.4. Water Vapor Transmission Rate

The water vapor transmission rate of fabricated PET/PA6 was analyzed using the gravimetric cup method with a fabric water vapor permeability measuring instrument (YG (B)216-II, Wenzhou Darong Textile Instrument Co., Ltd., Wenzhou, China) in accordance with the GB/T 12704.2-2009 standard. The constant temperature was 38 ± 2 °C, and the relative humidity was 50% ± 2%.

#### 3.3.5. Water Absorption Rate

The water absorption rate of fabricated PET/PA6 was estimated according to the GB/T 21655.1-2008 standard.

#### 3.3.6. Liquid Wicking Height and Liquid Wicking Rate

The liquid wicking height and liquid wicking rate of fabricated PET/PA6 were analyzed in accordance with the FZ/T 01071-2008 standard. The test time was 5 min. Three specimens (with dimensions of 250 mm × 30 mm) were measured per sample, and the average value was reported. 

#### 3.3.7. Spray Rating

The spray rating of fabricated PET/PA6 was characterized according to the GB/T 4745-1997 standard. 

#### 3.3.8. Tensile Strength and Elongation at Break

The tensile strength and elongation at break of fabricated PET/PA6 were determined using a tensile tester (Instron 3369, America Instron Co., Ltd., Boston, USA) in accordance with the GB/T 24218.3-2010 standard, with a crosshead speed of 100 mm/min. The specimen width was 50 mm ± 0.5 mm, and the specimen length was 250 mm ± 0.5 mm. The clamping distance was 200 mm. The same specimens were tested five times to obtain the average value. 

#### 3.3.9. Softness

The softness of fabricated PET/PA6 was measured using a softness tester (YN-L-051, Dongguan YP Testing Equipment Co., Ltd., Dongguan, China) in accordance with the GB/T 8942-2016 standard. The size of tested samples was 100 mm × 100 mm, and the width of the selected gap was 10 mm. The speed of the probe was 1.20 mm/s ± 0.24 mm/s. At least five measurements were averaged to obtain a reliable value. 

#### 3.3.10. Crease Recovery Property

The crease recovery angle of PET/PA6 was measured using a Digitel wrinkle recovery tester (YG541L, Quanzhou Meibang instrument Co., Ltd., Quanzhou, China) according to the QB/T 3819-1997 standard. The sample size was 40 mm × 15 mm. The load weight was 10 N. The compression area was 20 mm × 15 mm. The compression time was 300 s ± 5 s. The crease recovery angles were tested after 300 s of pressure relief. Twenty specimens of each sample were analyzed (10 each in transverse and longitudinal directions) [26]. 

## 4. Conclusions

The PET/PA6 hollow segmented pie microfiber nonwovens were prepared using bicomponent spunbond hydro-entangled technology and modified by UV/TiO_2_/H_2_O_2_. The influence of H_2_O_2_, nano-TiO_2_, and UV irradiation time on the structure and properties of PET/PA6 were analyzed. The main findings of the paper are as follows: 

The surface roughness of microfibers treated by UV/H_2_O_2_ slightly increased. When TiO_2_ was introduced, the surface of modified microfibers was coated with a layer of granular material. In addition, EDS data revealed that the weight concentration and atomic concentration of Ti increased with the addition of TiO_2_ and the increase in UV irradiation time, reaching 44.97% and 19.06%, respectively. Furthermore, the elemental composition and stoichiometry were determined using XPS analysis. It was found that the C 1*s* peak for the sample modified by UV/TiO_2_/H_2_O_2_ could be deconvoluted into six components: C–C–C, C–C–O, O–C=O, N–C=O, N–C–C, and C–C=O. A mechanism was proposed for the hydrophilic modification of PET/PA6 by UV/TiO_2_/H_2_O_2_. 

The contact angle, water vapor transmission rate, water absorption, capillary effect, and waterproof performance were used to characterize the hydrophilic properties of PET/PA6. The results demonstrated that the water contact angle of PET/PA6 modified with TiO_2_/H_2_O_2_ and 90 min UV irradiation was reduced to zero in 0.015 s. The water vapor transmission rate and the water absorption could reach 5567.49 g/(m^2^·24 h) and 438.81%, respectively. The liquid wicking height and liquid wicking rate of the nonwovens further increased, reaching 141.87 mm and 28.37 mm/min, respectively. In addition, the PET/PA6 had excellent waterproof performance with a spray rating of level 1.

The difference in tensile strength for PET/PA6 before and after modification was not significant, whereas the softness of modified PET/PA6 increased significantly. 

## Figures and Tables

**Figure 1 molecules-28-03826-f001:**
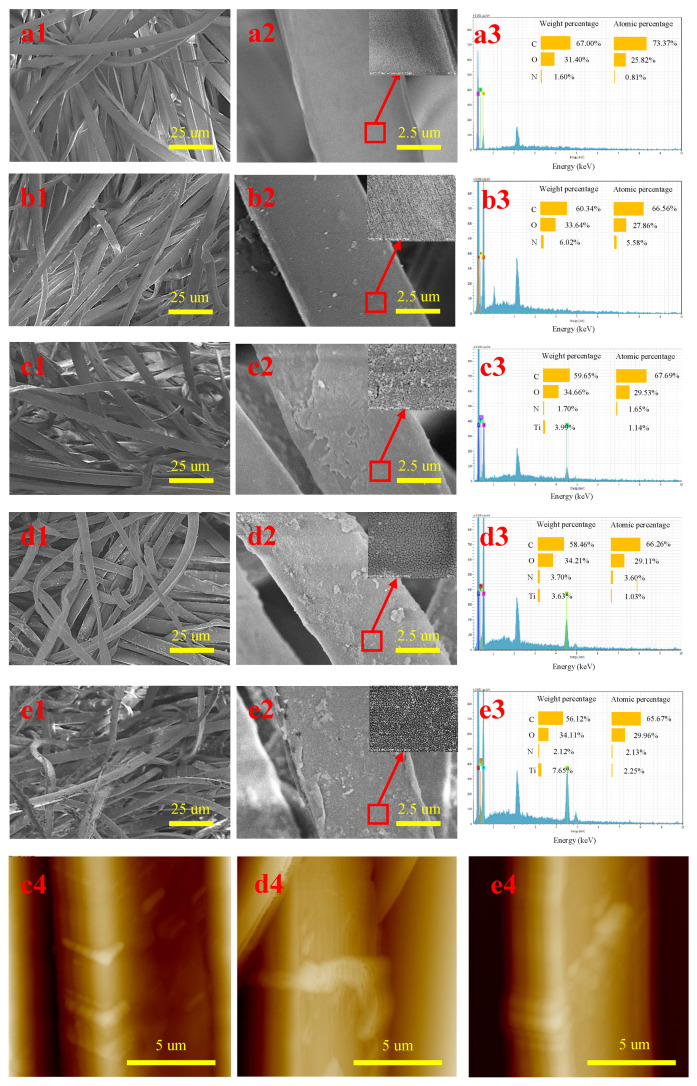
SEM microphotographs (1 and 2), EDS spectrum (3), and AFM images (4) of PET/PA6 before modification (**a**) and after modification by (**b**) H_2_O_2_/UV 45 min, (**c**) TiO_2_/UV 45 min, (**d**) TiO_2_/H_2_O_2_/UV 45 min, and (**e**) TiO_2_/H_2_O_2_/UV 90 min.

**Figure 2 molecules-28-03826-f002:**
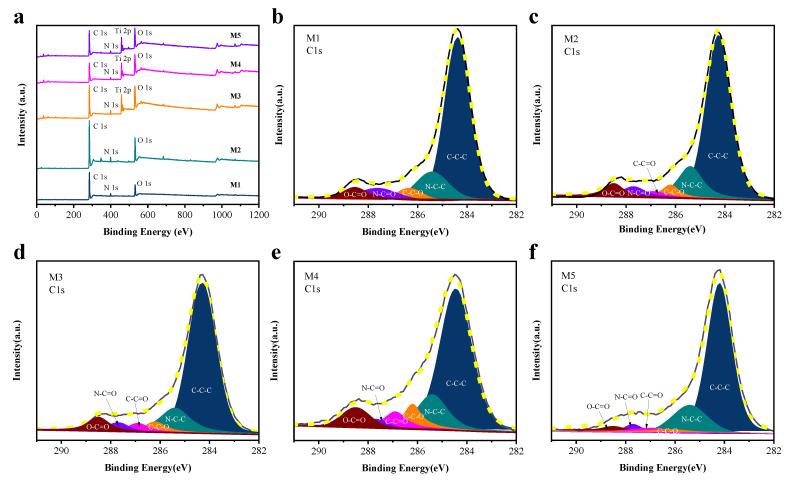
The wide survey scan of XPS spectra in the range 0–1200 eV (**a**); high-energy-resolution XPS spectra of the C 1*s* region: PET/PA6 before modification (**b**) and after modification by (**c**) H_2_O_2_/UV 45 min, (**d**) TiO_2_/UV 45 min, (**e**) TiO_2_/H_2_O_2_/UV 45 min, (**f**) TiO_2_/H_2_O_2_/UV 90 min.

**Figure 3 molecules-28-03826-f003:**
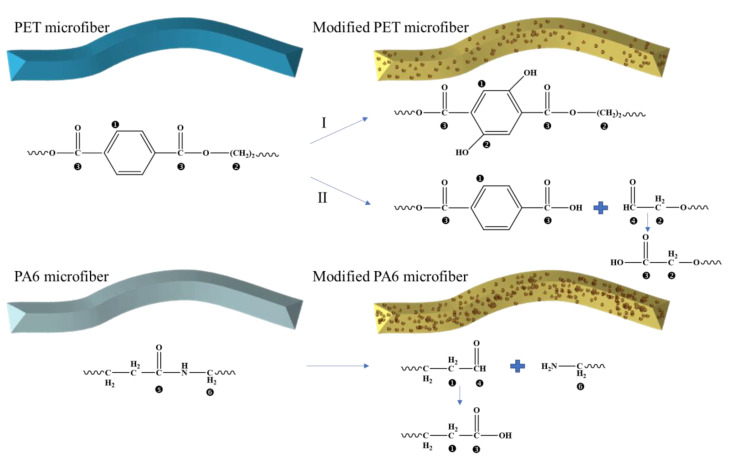
Proposed mechanism for hydrophilic modification of PET/PA6 by UV/TiO_2_/H_2_O_2_.

**Figure 4 molecules-28-03826-f004:**
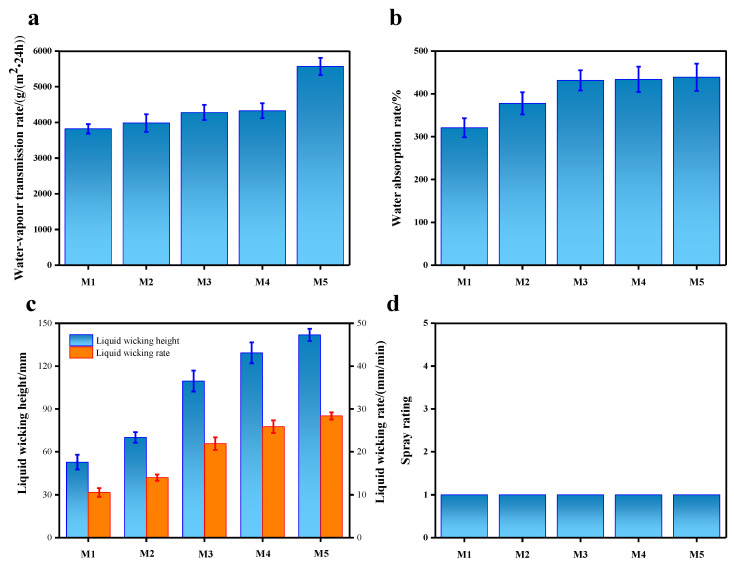
The hydrophilic properties of PET/PA6: (**a**) water vapor transmission rate; (**b**) water absorption rate; (**c**) liquid wicking height and liquid wicking rate; (**d**) spray rating.

**Figure 5 molecules-28-03826-f005:**
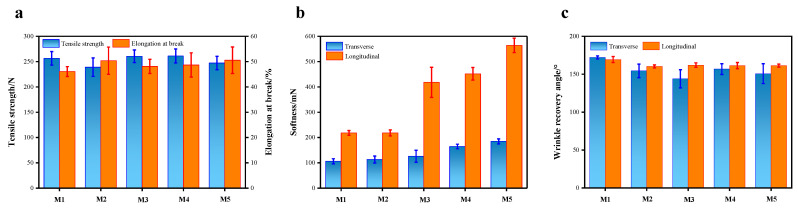
The mechanical properties of PET/PA6: (**a**) tensile strength and elongation at break; (**b**) softness; (**c**) crease recovery angle.

**Figure 6 molecules-28-03826-f006:**
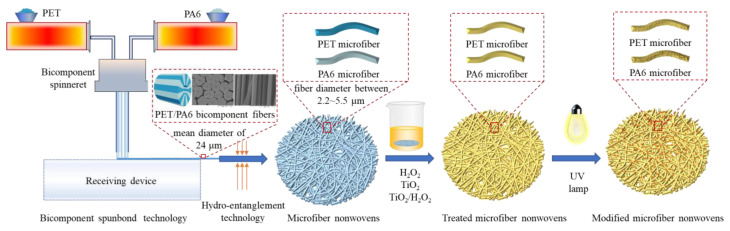
Schematic illustration of the preparation and modification process of PET/PA6 hollow segmented pie microfiber nonwovens using bicomponent spunbond hydro-entangled technology and UV-induced photodegradation.

**Table 1 molecules-28-03826-t001:** The surface chemical composition of PET/PA6 from XPS analysis.

Sample		M1	M2	M3	M4	M5
Atomic composition (%)	C 1*s*	78.80	75.56	60.00	59.58	55.50
	O 1*s*	16.60	19.91	28.25	28.35	31.42
	N 1*s*	4.60	4.53	4.28	3.84	3.74
	Ti 2*p*	0	0	7.47	8.23	9.34
High energy resolution of C1s (%)	C–C–C ^❶^	67.56	67.38	67.27	62.20	60.50
	C–C–O ^❷^	4.54	4.56	4.71	7.87	8.51
	O–C=O ^❸^	4.99	5.04	6.00	6.53	7.65
	C–C=O ^❹^	0	2.81	3.63	5.71	6.04
	N–C=O ^❺^	7.64	4.84	2.98	2.27	1.95
	N–C–C ^❻^	15.28	15.38	15.40	15.42	15.36

^❶–❻^ The number of the chemical bonds.

**Table 2 molecules-28-03826-t002:** The contact angles and droplet disappearance times of PET/PA6.

Sample	M1	M2	M3	M4	M5
Contact angle (at 0 s, °)	137.718	136.734	134.637	134.637	130.775
Contact angle (at 0.015 s, °)	137.062	136.389	133.150	132.769	0
Droplet disappearance time (s)	27.465	1.680	0.271	0.072	0.015

**Table 3 molecules-28-03826-t003:** The impregnating solution and UV irradiation time.

Sample	Impregnating Solution	Composition	UV Irradiation Time (min)
M1	/	/	/
M2	H_2_O_2_	2.4 mL of H_2_O_2_ solution and 180 mL of water	45
M3	TiO_2_	20 mL of nano-TiO_2_ sol and 160 mL of water	45
M4	TiO_2_/H_2_O_2_	2.4 mL of H_2_O_2_ solution, 20 mL of nano-TiO_2_ sol, and 160 mL of water	45
M5	TiO_2_/H_2_O_2_	2.4 mL of H_2_O_2_ solution, 20 mL of nano-TiO_2_ sol and 160 mL of water	90

## Data Availability

The data that support the findings of this study are available from the corresponding author (B.Z.) upon reasonable request.

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
