# Peer review of "Hydrophilic Modification of Polyester/Polyamide 6 Hollow Segmented Pie Microfiber Nonwovens by UV/TiO2/H2O2"

_molecules, 2023, doi:10.3390/molecules28093826_

Round 1

Reviewer 1 Report

Abstract

- Add one sentence with the purpose/aim/application - why the authors thought there was a need for study like this.

- Line 13: The abbreviation PET/PA6 HSBSHMN should be just PET/PA6 for easiest reading, since you already explain what kind of nonwoven this is. Please change it also through all the text.

- Line 25: g/(m2·24h) – change this unit with proper SI Units (International System of Units). Please change it also through all the text.

Introduction

- Last paragraph: add a short discussion on why the authors thought there was a need for study like this. The originality as well as the importance of the study should be emphasized.

Results and discussion

- In general, the Results and discussion should be compared with the results of other literature sources.

- Line 90-91: This may be due to the oxidation of chlorine peroxide to generate new oxides on the sur-face of the insulation, or it may be some small particulate contaminants adhering to it. Really? From where chlorine peroxide came? What about the formation of radicals, which could destroy the surface? Or something else? Please find the explanation of other study(ies) and add the reference(s).

- Line 106-107: Moreover, by increasing the UV irradiation time, the weight concentration and atomic concentration of Ti further increased to 44.97% and 19.06%. Why? Explain/compare with other studies/references.

- Figure 1: Enlarge individual micrographs and EDS spectrum to be more readable – you can turn Figure in horizontal direction.

- All Figures and Tables should be inserted into the main text close to their first citation.

- Line 122-124: Why Ti increased, and C decreased? Explain.

- Line 125-133: Why deconvolution is used in this study and what these results mean (degradation? Transformation? TiO2 coating? Or something else? – detail explanation is needed to understand the mechanism behind the modification.

- Line 140-147 should relate to previous explanation above (this explanation in present form is not connected with any of your results).   

- Why N is not visible in EDS, although on XPS is?

- Line 155: of PET/PA6 – delete of

- Line 158-162: Rephrase the sentence to be clearer (split into 2 sentences).

-Line 192: What means spray rating of level 1 – explain.

- Line 204-205: It is because the nano-TiO2 sol can be used as a UV shielding agent and has a protective effect on the internal structure of fibres. Add the reference.

-Line 212-213: The mechanical properties of the nonwovens before and after the modification met the demand of nonwoven textiles for synthetic leather. And this is? Add the reference.

Materials and Methods

- Line 231-234: Instead of: Secondly, the filament microfiber nonwovens were washed by ethanol and then immersed in solutions for 30 min, such 232 as H2O2 aqueous solution (2.4 ml H2O2 solution and 180 ml water), nano-TiO2 hydrosol 233 (20 mL nano-TiO2 sol and 160 mL water), nano-TiO2/H2O2 hydrosol (2.4 mL H2O2 solution, 234 20 mL nano-TiO2 sol and 160 mL water). Should be: Secondly, the filament microfiber nonwovens were washed by ethanol and then immersed in individual solution (Table 3). In Table 3 put detail description of individual solution for (+ add the time).

- Line 239: dry in an oven - at what temperature/time?

- Line 239-240: Last sentence should be deleted.

- Figure 6: Letters are too small, and thus, not readable in printed version – please correct the Figure.

- Line 251: Instead produce should be enhance or increase

- Line 254: The surface chemical composition of fabricated PET/PA6 HSBSHMN by X-ray photoelectron spectroscopy (XPS, Thermo Scientific K-Alpha, UK). The verb is missing.

- Line 259, 267, 271, etc.: the word standard is missing before or after DB44…, GB/T…

- Line 274: were 250x30 mm - in size… is missing

References

- More than 50% of References are more than 6 years old, therefore, it is a need for exchanging at least some of them with new ones. Exceptions are books or manuscripts on which the work is based on. It is better to have fewer references that out-of-date ones; and consecutively, the Introduction section should be updated.

Author Response

Response to Reviewer 1 Comments

Point 1: Abstract: Add one sentence with the purpose/aim/application - why the authors thought there was a need for study like this.

Response 1: The sentence has been supplemented in abstract part in the revised manuscript molecules-2286730.R1 as follows:

“In order to enhance the hydrophilic performance of PET/PA6, many methods have been applied, but the effects are not obvious. Ultraviolet (UV) irradiation treatment has been proved to be an effective method to improve the hydrophilicity of fabrics.”

Point 2:  Abstract: Line 13: The abbreviation PET/PA6 HSBSHMN should be just PET/PA6 for easiest reading, since you already explain what kind of nonwoven this is. Please change it also through all the text.

Response 2: Thanks for the referee’s suggestion. We have carefully examined all the text, and changed PET/PA6 HSBSHMN to PET/PA6.

Point 3: Abstract: Line 25: g/(m2·24h) – change this unit with proper SI Units (International System of Units). Please change it also through all the text.

Response 3: The water-vapour transmission rate of PET/PA6 was tested in accordance with the standard GB/T 12704.2-2009: Textiles-Test method for water-vapour transmission of fabrics-Part 2: Water method. According to the standard GB/T 12704.2-2009, the g/(m2·24h) is the unit of water-vapour transmission rate. Duo et al [1,2] also used the g/(m2·24h) as the unit of water-vapour transmission rate.  

  1. Duo, Y.; Qian, X.; Zhao, B.; Gao, L.; Guo, X.; Zhang, S.; Bai, H.; Tang, L. Easily Splittable Hollow Segmented-Pie Microfiber Nonwoven Material with Excellent Filtration and Thermal-Wet Comfort for Energy Savings. J. Mater. Res. Technol.-Jmrt 2022, 17, 876–887.
  2. Duo, Y.; Qian, X.; Zhao, B.; Qian, Y.; Xu, P. Improving Hygiene Performance of Microfiber Synthetic Leather Base by Mixing Polyhydroxybutyrate Nanofiber. J. Eng. Fibers Fabr. 2019, 14.

Point 4: Introduction: Last paragraph: add a short discussion on why the authors thought there was a need for study like this. The originality as well as the importance of the study should be emphasized.

Response 4: The discussion has been supplemented in last paragraph in the revised manuscript molecules-2286730.R1 as follows:

“The influences of H2O2, nano-TiO2 and UV irradiation time on the performances of PET/PA6 were systematically investigated. This study had the goal of developing a method for Polyester/Polyamide 6 hollow segmented pie microfiber nonwovens to assist in solving the current hydrophobicity problem. These results are also useful in future hydrophilic modification of filament microfiber nonwovens.”

Point 5:  Results and discussion: In general, the results and discussion should be compared with the results of other literature sources.

Response 5: Thanks for the referee’s suggestion. It has been revised in the revised manuscript molecules-2286730.R1.

Point 6: Results and discussion: Line 90-91: This may be due to the oxidation of chlorine peroxide to generate new oxides on the surface of the insulation, or it may be some small particulate contaminants adhering to it. Really? From where chlorine peroxide came? What about the formation of radicals, which could destroy the surface? Or something else? Please find the explanation of other study(ies) and add the reference(s).

Response 6: The explanation and reference have been supplemented in the revised manuscript molecules-2286730.R1 as follows:

“Some bright spots can be seen under high magnifications. This may be due to the reaction of H2O2 and amide bond generate the cracking of PA6 macromolecular chain, or it may be the flaws generated during manufacturing[15,16].”

Point 7:  Results and discussion: Line 106-107: Moreover, by increasing the UV irradiation time, the weight concentration and atomic concentration of Ti further increased to 44.97% and 19.06%. Why? Explain/compare with other studies/ references.

Response 7: The explanation and reference have been supplemented in the revised manuscript molecules-2286730.R1 as follows:

“TiO2 network were formed on the surface of filaments via the interaction of the TiO2 sol with OH groups[17]. Here, with increasing UV irradiation time, the accumulation on the TiO2 layer increased, that is, the amounts of the TiO2 particles increased[18]. Figure 1 also shows the AFM images of specimen M3, M4 and M5. The roughness was 339 nm, 641 nm and 1007 nm, which indicated the surfaces became much rougher ascribing to the deposited TiO2 layer.”

Point 8: Results and discussion: Figure 1: Enlarge individual micrographs and EDS spectrum to be more readable – you can turn Figure in horizontal direction. All Figures and Tables should be inserted into the main text close to their first citation.

Response 8: Thanks for the referee’s suggestion. We have carefully examined and turned Figure 1 in horizontal direction in the revised manuscript molecules- 2286730.R1.  

Point 9:  Results and discussion: Line 122-124: Why Ti increased, and C decreased? Explain.

Response 9: The explanation has been supplemented in the revised manuscript molecules-2286730.R1 as follows:

“This is mainly because more and more TiO2 have been introduced to the microfibre's surface.”

Point 10:  Results and discussion: Line 125-133: Why deconvolution is used in this study and what these results mean (degradation? Transformation? TiO2 coating? Or something else? – detail explanation is needed to understand the mechanism behind the modification.

Results and discussion: Line 140-147 should relate to previous explanation above (this explanation in present form is not connected with any of your results). 

Response 10: The explanation has been supplemented in the revised manuscript molecules-2286730.R1 as follows:

“The amounts of carbon atoms in the different bonding environments is shown in Table 1. It can be seen that the amount of C-C-C and N-C=O observed will decrease, while C-C-O, O-C=O, C-C=O and N-C-C observed will increase. Furthermore, the amount of C-C=O increase from 0 to 6.04%, which indicate aldehyde formation due to chain scission[20].”

Point 11: Results and discussion: Why N is not visible in EDS, although on XPS is?

Response 11: EDS has been retested and its data are supplemented in Figure 1 in the revised manuscript molecules-2286730.R1.  

Point 12:  Results and discussion: Line 155: of PET/PA6 – delete of.

Response 12: It has been revised in the revised manuscript molecules-2286730.R1.

Point 13: Results and discussion: Line 158-162: Rephrase the sentence to be clearer (split into 2 sentences).

Response 13: The sentence has been split into 2 sentences in the revised manuscript molecules-2286730.R1 as follows:

“The PET/PA6 were impregnated with H2O2 aqueous solution (M2), nano-TiO2 hydrosol (M3) and nano-TiO2/H2O2 hydrosol (M4), and then irradiated by UV lamp for 45 min. The contact angles of M2, M3 and M4 at the first contact with water droplets were 136.734°, 134.637° and 134.637°, and the contact angles of M2, M3 and M4 at 0.015 s contact were 136.389°, 133.150° and 132.769°, respectively.”

Point 14:  Results and discussion: Line 192: What means spray rating of level 1 – explain.

Response 14: The GB/T 4745-1997 descriptive rating scale corresponds to the ISO and AATCC photographic scale as follows:

GB 1=ISO 1=AATCC 50

GB 2=ISO 2=AATCC 70

GB 3=ISO 3=AATCC 80

GB 4=ISO 4=AATCC 90

GB 5=ISO 5=AATCC 100

Therefore, spray rating of level 1 means complete wetting of whole of upper surface.

Point 15:  Results and discussion: Line 204-205: It is because the nano-TiO2 sol can be used as a UV shielding agent and has a protective effect on the internal structure of fibres. Add the reference.

Results and discussion: Line 212-213: The mechanical properties of the nonwovens before and after the modification met the demand of nonwoven textiles for synthetic leather. And this is? Add the reference.

Response 15: Based upon the above comments from the reviewer, it has been revised in the revised manuscript TRJ-18-0066.R1.

Point 16:  Materials and Methods: Line 231-234: Instead of: Secondly, the filament microfiber nonwovens were washed by ethanol and then immersed in solutions for 30 min, such as H2O2 aqueous solution (2.4 ml H2O2 solution and 180 ml water), nano-TiO2 hydrosol (20 mL nano-TiO2 sol and 160 mL water), nano-TiO2/H2O2 hydrosol (2.4 mL H2O2 solution, 20 mL nano-TiO2 sol and 160 mL water). Should be: Secondly, the filament microfiber nonwovens were washed by ethanol and then immersed in individual solution (Table 3). In Table 3 put detail description of individual solution for (+ add the time).

Response 16: Based upon the above comments from the reviewer, it has been revised in the revised manuscript molecules-2286730.R1 as follows:

“Secondly, the filament microfiber nonwovens were washed by ethanol and then immersed in individual impregnating solution. The detail description of impregnating solution and UV irradiation time are listed in Table 3.”

Point 17:  Materials and Methods: Line 239: dry in an oven - at what temperature/time?

Response 17: The temperature and time have been supplemented in the revised manuscript molecules-2286730.R1 as follows:

“Finally, the samples were dried in an oven for 2 h under high-temperature (80ºÐ¡) to obtain the modified PET/PA6.”

Point 18: Materials and Methods: Line 239-240: Last sentence should be deleted.

Response 18: Thanks for the referee’s suggestion. The last sentence has been deleted in the revised manuscript molecules-2286730.R1.

Point 19:  Materials and Methods: Figure 6: Letters are too small, and thus, not readable in printed version – please correct the Figure.

Response 19: Thanks for the referee’s suggestion. It has been revised in the revised manuscript molecules-2286730.R1.

Point 20:  Materials and Methods: Line 251: Instead produce should be enhance or increase

Response 20: Thanks for the referee’s suggestion. It has been revised in the revised manuscript molecules-2286730.R1.

Point 21: Materials and Methods: Line 254: The surface chemical composition of fabricated PET/PA6 HSBSHMN by X-ray photoelectron spectroscopy (XPS, Thermo Scientific K-Alpha, UK). The verb is missing.

Response 21: Thanks for the referee’s suggestion. The verb has been supplemented in the revised manuscript molecules-2286730.R1 as follows:

“The surface chemical composition of fabricated PET/PA6 was tested by X-ray photoelectron spectroscopy (XPS, Thermo Scientific K-Alpha, UK).”

Point 22:  Materials and Methods: Line 259, 267, 271, etc.: the word standard is missing before or after DB44…, GB/T…

Response 22: Thanks for the referee’s suggestion. It has been revised in the revised manuscript molecules-2286730.R1.

Point 23: Materials and Methods: Line 274: were 250x30 mm - in size… is missing

Response 23: It has been revised in the revised manuscript molecules-2286730.R1 as follows:

“Three specimens (with dimensions of 250 mm×30 mm) were measured per sample and the average value was reported.”

Point 24:  References: More than 50% of References are more than 6 years old, therefore, it is a need for exchanging at least some of them with new ones. Exceptions are books or manuscripts on which the work is based on. It is better to have fewer references that out-of-date ones; and consecutively, the Introduction section should be updated.

Response 24: Thanks for the referee’s suggestion. It has been revised in the revised manuscript molecules-2286730.R1.

According to referee’s advice, this manuscript was edited for proper English language, grammar, punctuation, spelling, and overall style.

Finally, the authors greatly appreciate the above helpful comments and constructive suggestions from the reviewer.

Reviewer 2 Report

This manuscript pertains to the current topic, namely the development of hydrophilic fibers. The results are interesting! However, I find several aspects are not thoroughly discussed and therefore should be included to further improve the quality of the manuscript. Please consider the following revisions.

1. What is the surface roughness of 'as fabricated' fibers and modified fibers (Fig. 3)? Please add Atomic Force Microscopy (AFM) data.

2. Please add cross-sectional scanning electron microscope (SEM) image of as fabricated hollow fibers.

3. What is the effect of fiber surface modification on elastic recovery of fibers? Please add data on elastic recovery of 'as fabricated' and modified fibers.

4. Please add information on thickness of TiO2 layer, surface roughness, fiber diameter and fiber's Young's modulus to Table 3.

5. What is the accuracy of tensile measurements? Please add stress - strain curves to next version of manuscript.

6. The authors mention increasing softness of nonwovens (p. 7). What is the reason for increasing material softness with increasing nanoparticles content?

7. Please add explanation to Fig. 3 in details.

Author Response

Response to Reviewer 2 Comments

Point 1:  What is the surface roughness of 'as fabricated' fibers and modified fibers (Fig. 3)? Please add Atomic Force Microscopy (AFM) data.

Response 1: Thanks for the referee’s suggestion. The AFM date has been supplemented in Figure 1 in the revised manuscript molecules-2286730.R1 as follows:

“TiO2 network were formed on the surface of filaments via the interaction of the TiO2 sol with OH groups[17]. Here, with increasing UV irradiation time, the accumulation on the TiO2 layer increased, that is, the amounts of the TiO2 particles increased[18]. Figure 1 also shows the AFM images of specimen M3, M4 and M5. The roughness was 339 nm, 641 nm and 1007 nm, which indicated the surfaces became much rougher ascribing to the deposited TiO2 layer.”

Point 2:  Please add cross-sectional scanning electron microscope (SEM) image of as fabricated hollow fibers.

Response 2: Thanks for the referee’s suggestion. The cross-sectional SEM image of as fabricated hollow fibers has been supplemented in Figure 6 in the revised manuscript molecules-2286730.R1 as follows:

Point 3: What is the effect of fiber surface modification on elastic recovery of fibers? Please add data on elastic recovery of 'as fabricated' and modified fibers.

Response 3: Based upon the above comments from the reviewer, the crease recovery angle of microfiber nonwovens has been tested and the data has been supplemented in Figure 5 in the revised manuscript molecules-2286730.R1 as follows:

“3.3.10. Crease recovery property

The crease recovery angle of PET/PA6 was measured with a digitel wrinkle recovery tester (YG541L, Quanzhou Meibang instrument Co., Ltd, China) according to the QB/T 3819-1997 standard. The sample size was 40 mmÍ15 mm. The load weight was 10 N. The compression area was 20 mmÍ15 mm. The compression time was 300 s ± 5 s. The crease recovery angles were tested after 300 s of pressure relief. The number of each sample was 20, namely 10 samples of transverse and longitudinal direc-tions[26].”

“Meanwhile, As shown in Figure 5c, the PET/PA6 before and after modification have different crease recovery angle. The crease recovery property of unmodified nonwoven was greater than those of modified nonwovens. Modified nonwovens was easily wrinkled, and recovered flatness after being wrinkled was difficult.”

Point 4: Please add information on thickness of TiO2 layer, surface roughness, fiber diameter and fiber's Young's modulus to Table 3.

Response 4: Based upon the above comments from the reviewer, the fiber diameter has been supplemented in morphology analysis in the revised manuscript molecules-2286730.R1 as follows:

“Figure 1 shows the SEM images of Fabricated PET/PA6 and its surface EDS spec-trum. As shown in Figure 1a1 and Figure 1a2, it can be seen that the surface of untreated microfiber is relatively smooth and shiny. The microfiber diameter was 5.89 um. The surface roughness of the microfibers after modification with UV/H2O2 slightly increased (see Figure 1b1 and Figure 1b2). Some bright spots can be seen under high magnifications. This may be due to the reaction of H2O2 and amide bond generate the cracking of PA6 macromolecular chain, or it may be the flaws generated during man-ufacturing[15,16]. When TiO2 was introduced, the modified microfibers were coated with a layer of granular material on the surface (see Figure 1(c1, d1 and e1) and Figure 1(c2, d2 and e2)), and the materials failed to wash off after repeated washing. The microfiber diameter was 6.16 um, 6.23 um and 6.55 um, respectively. High magnification electron micrographs showed that many irregular fine particles were distributed on the surface of the treated microfibers. The partial agglomeration of particles causes some large micron-sized particles to adhere to the fiber surface.”

The thickness of TiO2 layer depends on the fiber diameter.

The surface roughness has been supplemented in Figure 1.

The microfiber can not be separated from the fabricated nonwovens. Therefore, fiber's Young's modulus can not be measured.

Point 5:  What is the accuracy of tensile measurements? Please add stress-strain curves to next version of manuscript.

Response 5: Thanks for the referee’s suggestion. In order to ensure the accuracy of tensile strength and elongation at break, the same group of specimens were tested 5 times to obtain the average value. Many previous references[1,2] analyzed tensile data in the same way.

  1. Duo, Y.; Qian, X.; Zhao, B.; Qian, Y.; Xu, P. Improving Hygiene Performance of Microfiber Synthetic Leather Base by Mixing Polyhydroxybutyrate Nanofiber. J. Eng. Fibers Fabr. 2019, 14.
  2. Zhao, B.; Qian, X.; Qian, Y.; Fan, J.; Feng, Y.; Duo, Y.; Zhang, H. Preparation of High-Performance Microfiber Synthetic Leather Base Using Thermoplastic Polyurethane/Sulfonated Polysulfone Electrospun Nanofibers. Text. Res. J. 2019, 89, 2813–2820.

Point 6:  The authors mention increasing softness of nonwovens (p.7). What is the reason for increasing material softness with increasing nanoparticles content?

Response 6: The higher the value of softness, the worse the softness of nonwovens. Therefore, the softness property decreased with increasing nanoparticles content.

Point 7:  Please add explanation to Fig. 3 in details.

Response 7: The explanation has been supplemented in the revised manuscript molecules-2286730.R1 as follows:

“The amounts of carbon atoms in the different bonding environments is shown in Table 1. It can be seen that the amount of C-C-C and N-C=O observed will decrease, while C-C-O, O-C=O, C-C=O and N-C-C observed will increase. Furthermore, the amount of C-C=O increase from 0 to 6.04%, which indicate aldehyde formation due to chain scission[20].”

Finally, the authors greatly appreciate the above helpful comments and constructive suggestions from the reviewer.

Round 2

Reviewer 1 Report

- Figure 1: add scale bar on individual SEM (although presented, but not clearly visible) and AFM image .  

Author Response

Point 1: add scale bar on individual SEM (although presented, but not clearly visible) and AFM image.

Response 1: Thanks for the referee’s suggestion. The scale bar has been supplemented in Figure 1 in the revised manuscript molecules-2286730.R2 as follows:

Finally, the authors greatly appreciate the above helpful comments and constructive suggestions from the reviewer.
